# Expert–Novice Level Classification Using Graph Convolutional Network Introducing Confidence-Aware Node-Level Attention Mechanism

**DOI:** 10.3390/s24103033

**Published:** 2024-05-10

**Authors:** Tatsuki Seino, Naoki Saito, Takahiro Ogawa, Satoshi Asamizu, Miki Haseyama

**Affiliations:** 1Graduate School of Information Science and Technology, Hokkaido University, Sapporo 060-0814, Japan; seino@lmd.ist.hokudai.ac.jp; 2Office of Institutional Research, Hokkaido University, Sapporo 060-0808, Japan; saito@lmd.ist.hokudai.ac.jp; 3Faculty of Information Science and Technology, Hokkaido University, Sapporo 060-0814, Japan; ogawa@lmd.ist.hokudai.ac.jp; 4National Institute of Technology, Kushiro College, Kushiro 084-0916, Japan; asamizu@kushiro-ct.ac.jp

**Keywords:** expert–novice level classification, motion data, graph convolutional network, attention mechanism

## Abstract

In this study, we propose a classification method of expert–novice levels using a graph convolutional network (GCN) with a confidence-aware node-level attention mechanism. In classification using an attention mechanism, highlighted features may not be significant for accurate classification, thereby degrading classification performance. To address this issue, the proposed method introduces a confidence-aware node-level attention mechanism into a spatiotemporal attention GCN (STA-GCN) for the classification of expert–novice levels. Consequently, our method can contrast the attention value of each node on the basis of the confidence measure of the classification, which solves the problem of classification approaches using attention mechanisms and realizes accurate classification. Furthermore, because the expert–novice levels have ordinalities, using a classification model that considers ordinalities improves the classification performance. The proposed method involves a model that minimizes a loss function that considers the ordinalities of classes to be classified. By implementing the above approaches, the expert–novice level classification performance is improved.

## 1. Introduction

In the context of sports, the transfer of “expert techniques” from outstanding athletes and coaches to the next generation of players is essential for development. However, most expert techniques are tacit knowledge, and the transfer of such techniques requires prolonged guidance from experienced athletes or coaches. Thus, the construction of support technologies to facilitate the efficient transfer of these expert techniques is expected. To effectively implement support technologies, it is essential to delineate the differences between expert and novice athletes. Therefore, the classification of athletes into “expert” and “novice” is a fundamental methodology [1]. In recent years, the popularization of wearable devices, such as smartwatches and motion capture devices, has facilitated the acquisition of biometric data, and various methods for expert–novice level classification using biometric data have been proposed [2,3,4,5,6,7,8,9,10,11,12,13,14,15,16,17,18,19,20]. For example, Kuo et al. proposed a classification method for laparoscopic surgical skills based on multiple machine learning methods such as a multilayer perceptron using gaze information [19]. Furthermore, Guo et al. proposed a skill-level classification method based on convolutional neural networks using a Single Inertial Sensor attached to the arm [20]. In particular, motion data are closely related to tacit knowledge. Several expert–novice level classification methods using motion data have been proposed [21,22,23,24,25,26,27]. For example, Ross et al. proposed a method for classifying athletes’ skill levels using machine learning techniques such as support vector machines and logistics regression based on motion data collected during specific movements [25]. Furthermore, Vincenzo et al. proposed a method for classifying violin performance levels by the random forest method using motion data [26]. In addition, Xuan et al. proposed a method for classifying surgical skill levels using a convolutional neural network (CNN) and long short-term memory (LSTM) based on motion data collected during surgical simulation [27]. From the above methods, expert–novice classification is realized using a machine-learning-based approach with biometric information. In particular, motion data have attracted attention as information that can accurately classify expert–novice levels and are used in many previous methods.

In classification tasks using motion data, many methods handle motion as a graph structure, which is used to construct a graph convolutional network (GCN) [28] based on motion data [29,30,31,32,33,34,35,36,37,38]. GCNs allow the relationships between joints in the human body to be captured in a graph structure, facilitating the classification of complex movements. However, conventional GCNs, typically used in classification tasks, only output classification results without providing explanations. Therefore, there is a need for methods that elucidate the reasoning behind the classification results. In this regard, a classification method using motion data via a spatiotemporal attention GCN (STA-GCN), which introduces the attention mechanism into the GCN, has been proposed [39]. The STA-GCN improves classification performance and provides explanations for the classification results. In the STA-GCN, the feature extractor is placed close to the input, whereas the attention and perception branches are placed closer to the output. The attention branch performs classification using feature maps obtained using the feature extractor and generates attention nodes and edges. These generated attention nodes and edges are used to highlight the parts that are critical for accurate classification. Conversely, the perception branch performs the final classification using feature maps, attention nodes, and attention edges derived from both the feature extractor and the attention branch. From the above procedures, the attention mechanism in the STA-GCN enables the highlighting of important parts for classification. However, if the parts emphasized by the attention mechanism differ from the actual focal parts, there is a potential for reduced classification performance [40]. Therefore, in the attention mechanism, the influence of attention that fails to highlight important parts needs to be diminished.

To address this issue, we previously proposed an expert–novice level classification method (confidence-aware STA-GCN: ConfSTA-GCN) that introduces an attention mechanism that considers the confidence measure of critical parts [41]. Because the confidence measure is treated as the probability of class assignment obtained through expert–novice level classification in the attention branch, the same confidence measure is applied as a weight to all attention nodes in the previous method. However, it is anticipated that there will be variations in the confidence measure at each attention node for accurate classification. Therefore, by calculating different confidence measures for each attention node, classification performance can be improved. In addition, previous methods construct classification models under the assumption that there is no ordinality between the classes of the expert and novice levels; thus, they do not consider relationships between these classes. Given the ordinariness of the expert–novice levels, this can lead to limitations in classification performance.

In this study, we propose a method for expert–novice level classification using a GCN with a confidence-aware node-level attention mechanism. The proposed method calculates the probability of belonging to an actual expert–novice level when specific attention nodes are excluded. This process is repeated for the number of attention nodes, and the computed probabilities are regarded as confidence measures. A novel attention mechanism that considers the confidence measure of each attention node is one of the main contributions of this study. The perception branch outputs the final classification results using feature maps computed from these attention nodes, and these features are adjusted according to the confidence measure. Furthermore, the proposed method considers the ordinality of the expert–novice level, an aspect not considered in previous methods using attention mechanisms. Because this allows for consideration of the relationships between classes, it is expected to further improve of expert–novice level classification performance. The main contributions of this study can be summarized as follows.

Proposal of a method for improving existing GCN-based classification approaches by individually calculating and applying the confidence measure to attention nodes.Construction of a classification model that allows for consideration of the order of expert–novice levels among classes.

Note that this is an extended version of the ConfSTA-GCN for skeleton-based expert–novice level classification [41]. Specifically, the proposed method can calculate the confidence measure for each joint, separately, resulting in a node-level attention mechanism.

This paper is organized as follows. In Section 2, the classification of the expert–novice levels using the STA-GCN with a confidence-aware node-level attention mechanism is explained. The experimental results are described in Section 3 to evaluate the classification performance of our method. Finally, Section 4 concludes this study and describes future work.

## 2. Classification of Expert–Novice Levels Using STA-GCN with Confidence-Aware Node-Level Attention Mechanism

This section describes the proposed method to improve the existing approach and this study’s novelty. It also structures a classification model considering the expert–novice level ordering relationship between classes. An overview of the proposed method is shown in Figure 1. The proposed method comprises a feature extractor, an attention branch, and a perception branch. First, the proposed method uses a spatiotemporal graph (ST-graph) to represent spatial and temporal motion data as a graph structure. Using the feature extractor process, feature maps are calculated. Next, the attention branch obtains attention nodes, which represent the significance of each joint, and attention edges, which indicate the important relationships between joints. Furthermore, the attention branch uses the confidence-aware node-level attention mechanism to generate a new feature map in which important joints and their connections are emphasized for classification. Finally, by inputting the attention nodes and edges along with the feature map into the perception branch, the classification results of the expert–novice levels are obtained. During this process, the attention nodes computed using the attention branch are output as a visualization of important joints for expert–novice level classification.

### 2.1. ST-Graph Construction and Feature Extractor

This subsection describes the calculation of features that take into account spatiotemporal information from motion data. Specifically, we describe the construction approach of an ST-graph and the feature extractor method separately. First, the proposed method constructs an ST-graph from motion data in the same manner as [29]. Specifically, the ST-graph represents human joints as nodes $v_{f,n}(f=1,2,\cdots ,F;F$ denoting the number of frames, n=1,2,⋯,N;N representing the number of nodes), as shown in Figure 2. The ST-graph connects them with inter-frame and intra-body edges. The inter-frame edges connect the same joint across consecutive frames, i.e., the *f*-th and (f+1)-th frames in the motion data. Conversely, the intra-body edges connect nodes in the ST-graph according to the adjacency relationships of each joint in the human body.

The proposed method computes the feature map using a spatiotemporal graph convolutional (STGC)-block. The network configuration of the STGC-block is depicted in Figure 3. This block performs spatial graph convolution (S-GC) and temporal graph convolution (T-GC). Let $y\left(v_{f,n}\right)\in R^{D}$ (*D* denoting the dimension of node features) be the feature vector for the *n*-th node in the *f*-th frame. Our method defines the feature map obtained from the ST-graph as $Y_{\mathrm{in}}={\left[y\left(v_{f,1}\right),y\left(v_{f,2}\right),\cdots ,y\left(v_{f,N}\right)\right]}^{⊤}\in R^{N\times D}$. First, the output $Y_{\mathrm{out}}^{\mathrm{space}}\in R^{N\times N}$, which is obtained by applying S-GC to the feature map $Y_{\mathrm{in}}$, is computed as follows:(1)$$Y_{\mathrm{out}}^{\mathrm{space}}=\sum_{h=1}^{H}W_{h}^{\mathrm{Edge}}\circ \left(\Lambda_{h}^{-1/2}\left(A_{h}^{\mathrm{space}}+I\right)\Lambda_{h}^{-1/2}\right)Y_{\mathrm{in}}W_{h}^{\mathrm{Node}},$$
where $W_{h}^{\mathrm{Node}}$ and $W_{h}^{\mathrm{Edge}}$ (h=1,2,⋯,H;H denoting the number of adjacent nodes connected by intra-body edges) denote the weight matrices of the nodes and edges, respectively, and $A_{h}^{\mathrm{space}}$ denotes the adjacency matrix in the spatial direction. The symbol “∘” denotes the Hadamard product, and $I\in R^{N\times N}$ denotes the identity matrix. Furthermore, $\Lambda_{h}\in R^{N\times N}$ denotes a diagonal matrix whose diagonal elements are $\Lambda_{nn}=\sum_{i}\left(A_{ni}+I_{ni}\right)$.

The proposed method calculates the output $Y_{\mathrm{out}}^{\mathrm{time}}\in R^{F\times N\times D}$ using T-GC as follows:(2)$$Y_{\mathrm{out}}^{\mathrm{time}}=\left[\begin{array}{cccc} y^{\mathrm{time}}\left(v_{1,1}\right) & y^{\mathrm{time}}\left(v_{1,2}\right) & \cdots & y^{\mathrm{time}}\left(v_{1,N}\right)\\ y^{\mathrm{time}}\left(v_{2,1}\right) & y^{\mathrm{time}}\left(v_{2,2}\right) & \cdots & y^{\mathrm{time}}\left(v_{2,N}\right)\\ \vdots & \vdots & \ddots & \vdots \\ y^{\mathrm{time}}\left(v_{F,1}\right) & y^{\mathrm{time}}\left(v_{F,2}\right) & \cdots & y^{\mathrm{time}}\left(v_{F,N}\right) \end{array}\right]$$
(3)$$y^{\mathrm{time}}\left(v_{f,n}\right)=\sum_{\tau =-\lfloor \kappa /2\rfloor }^{\left\lfloor \kappa /2\right\rfloor }\alpha_{\tau }\circ x\left(v_{f-\tau ,n}\right)\in R^{D},$$
where κ denotes the size of the T-GC kernel and $\alpha_{\tau }\in R^{D}$ denotes the weight vector of T-GC. In the STGC-block, $Y_{\mathrm{out}}^{\mathrm{FE}}\in R^{F\times N\times D}$ is calculated using the network architecture shown in Figure 3. The STGC-block consists of S-GC, batch normalization [42], the ReLU activation function [43], T-GC, and Dropout [44], with a skip connection [45].

### 2.2. Attention Branch

This work aims to improve the attention branch in the existing GCN-based classification. The attention edges $E\left(Y_{\mathrm{out}}^{\mathrm{FE}}\right)\in R^{1\times N\times N}$ are derived by applying several 1 × 1 convolution layers [46] and global average pooling (GAP) [46] to the feature map $Y_{\mathrm{out}}^{\mathrm{FE}}$. This is followed by batch normalization and the application of the Tanh and ReLU functions to convert the values of non-important parts to zero. The attention edges $E(Y_{\mathrm{out}}^{\mathrm{FE}})$, which contain only the connections important for classification, are computed by employing several 1 × 1 convolution layers and GAP to the feature map $Y_{\mathrm{out}}^{\mathrm{FE}}$. Subsequently, batch normalization is performed, and the Tanh and ReLU activation functions are used to convert the values of non-essential parts to zero, thereby computing the attention edges.

Furthermore, the attention branch in the proposed method employs a process to obtain the attention nodes and edges using the feature map $Y_{\mathrm{out}}^{\mathrm{FE}}$ calculated in the previous subsection. The attention nodes $V\left(Y_{\mathrm{out}}^{\mathrm{FE}}\right)\in R^{1\times N\times N}$ are obtained by applying several 1 × 1 convolution layers, batch normalization, upsampling, and the sigmoid function to $Y_{\mathrm{out}}^{\mathrm{FE}}$. In the upsampling process, linear interpolation is performed so that the number of frames in the feature map after the 1 × 1 convolution processes and batch normalization in T-GC and the input feature map $Y_{\mathrm{out}}^{\mathrm{FE}}$ become the same. Using the computed attention nodes $V(Y_{\mathrm{out}}^{\mathrm{FE}})$, a new feature map $Y_{\mathrm{out}}^{\mathrm{AN}}\in R^{1\times F\times N}$, which emphasizes information about important parts for expert–novice level classification, is computed as follows:(4)$$Y_{\mathrm{out}}^{\mathrm{AN}}=V\left(Y_{\mathrm{out}}^{\mathrm{FE}}\right)Y_{\mathrm{out}}^{\mathrm{FE}}.$$

The proposed method uses these attention nodes and edges for expert–novice level classification.

Our method applies the confidence-aware node-level attention mechanism to the feature map $Y_{\mathrm{out}}^{\mathrm{AN}}$ to emphasize important nodes. We obtain a novel feature map $Y_{\mathrm{out}}^{\mathrm{CAN}}$ from the attention nodes. In the confidence-aware node-level attention mechanism, we first calculate the confidence measure of each attention node. The calculation approach for the confidence measure is depicted in Figure 4. The proposed method computes the probability of belonging to each expert–novice level via a network in the attention branch when one of the attention nodes is masked, i.e., the attention value of the target node is set to zero. Let $c_{n,f}$ be the probability value calculated when the *n*-th attention node in the *f*-th frame is masked. The proposed method derives the confidence measure ${\bar{c}}_{n,f}$ of the *n*-th attention node in the *f*-th frame as follows:(5)$${\bar{c}}_{n,f}=1-c_{n,f}.$$

In the confidence-aware node-level attention mechanism, we calculate the confidence measure for all attention nodes, and the feature maps $Y_{\mathrm{out}}^{\mathrm{CAN}}$ are calculated using the feature maps $Y_{\mathrm{out}}^{\mathrm{AN}}$ and the confidence measure, as shown in the following equations:(6)$$Y_{\mathrm{out}}^{\mathrm{CAN}}=C\circ Y_{\mathrm{out}}^{\mathrm{AN}},$$
(7)$$C=\left[\begin{array}{cccc} {\bar{c}}_{11} & {\bar{c}}_{12} & \cdots & {\bar{c}}_{1N}\\ {\bar{c}}_{21} & {\bar{c}}_{22} & \cdots & {\bar{c}}_{2N}\\ \vdots & \vdots & \ddots & \vdots \\ {\bar{c}}_{F1} & {\bar{c}}_{F2} & \cdots & {\bar{c}}_{FN} \end{array}\right]\in R^{1\times F\times N}.$$

Equation (Equation 6) enables the controlled influence of attention nodes calculated using the confidence-aware node-level attention mechanism, moderated by the confidence measure of each attention node. From the above, the proposed method obtains the attention edges $E(Y_{\mathrm{out}}^{\mathrm{FE}})$ and the feature maps $Y_{\mathrm{out}}^{\mathrm{CAN}}$ calculated from the attention nodes $V(Y_{\mathrm{out}}^{\mathrm{FE}})$ in the attention branch to realize accurate expert–novice level classification.

### 2.3. Perception Branch

This work aims to compute classification results from features acquired in the feature extraction and attention branch. The perception branch obtains the final expert–novice level classification results by using a new feature map $Y_{\mathrm{out}}$. First, the proposed method computes $Y_{\mathrm{out}}^{\mathrm{per}}\in R^{N\times N}$ using the graph convolution of the attention edge and the feature map $Y_{\mathrm{in}}$ obtained from the ST-graph as follows:(8)$$Y_{\mathrm{out}}^{\mathrm{per}}=\sum_{\varphi =1}^{\phi }\left(\Lambda_{\varphi }^{-1/2}\left(A_{\varphi }^{\mathrm{per}}+I\right)\Lambda_{\varphi }^{-1/2}\right)Y_{\mathrm{in}}W_{\varphi }^{\mathrm{per}},$$
where $A_{\varphi }^{\mathrm{per}}\in R^{N\times N}$ (φ=1,2,⋯,ϕ; ϕ denoting the number of attention edges) denotes a normalized adjacency matrix of the attention edges, and $W_{\varphi }^{\mathrm{per}}\in R^{D\times N}$ denotes the weight matrix. Furthermore, $\Lambda_{\varphi }\in R^{N\times N}$ denotes a diagonal matrix. The proposed method obtains the final feature map $Y_{\mathrm{out}}\in R^{N\times N}$ using $Y_{\mathrm{out}}^{\mathrm{space}}$ calculated using Equation (Equation 1) and $Y_{\mathrm{out}}^{\mathrm{per}}$ as follows:(9)$$Y_{\mathrm{out}}=Y_{\mathrm{out}}^{\mathrm{space}}+Y_{\mathrm{out}}^{\mathrm{per}}.$$

Using multiple STGC-blocks, GAP, a fully connected layer, and the softmax function, we calculate the probability of belonging to each expert–novice level class. The class with the highest probability is considered the final classification result in the proposed method.

### 2.4. Training Approach

The purpose of this work is to construct a GCN learning approach that considers ordinality. The proposed method learns the STA-GCN by minimizing the loss function $L_{\mathrm{total}}$, which is calculated on the basis of the probability of belonging to each expert–novice level class. Specifically, $L_{\mathrm{total}}$ is defined as follows:(10)$$L_{\mathrm{total}}=L_{\mathrm{att}}+L_{\mathrm{per}},$$
(11)$$\begin{array}{c} L_{\mathrm{att}}=-\sum_{m=1}^{M}q^{\left(m\right)}\log p_{\mathrm{att}}^{\left(m\right)}-\sum_{m=1}^{M}\left({\left|\mathrm{label}-m\right|}^{2}\left(1-\delta^{\left(m\right)}\right)\log\left(1-p_{\mathrm{att}}^{\left(m\right)}\right)\right), \end{array}$$
(12)$$\begin{array}{c} L_{\mathrm{per}}=-\sum_{m=1}^{M}q^{\left(m\right)}\log p_{\mathrm{per}}^{\left(m\right)}-\sum_{m=1}^{M}\left({\left|\mathrm{label}-m\right|}^{2}\left(1-\delta^{\left(m\right)}\right)\log\left(1-p_{\mathrm{per}}^{\left(m\right)}\right)\right), \end{array}$$
where *M* represents the number of classes corresponding to the expert–novice levels, and label∈{1,2,⋯,M} denotes the ground truth of the expert–novice levels. $p_{\mathrm{att}}^{\left(m\right)}$ and $p_{\mathrm{per}}^{\left(m\right)}$ denote the probabilities of belonging to the *m*-th class (m=1,2,⋯,M), as determined by the attention and perception branches, respectively. $\delta^{\left(m\right)}$ is defined as follows:(13)$$\delta^{\left(m\right)}=\left\{\begin{array}{cc} 1 & \mathrm{if}\;m=\mathrm{label},\\ 0 & \mathrm{otherwise}. \end{array}\right.$$

In the proposed method, the squared difference between the ground truth and the classification result is used as a weight in the second term of Equations (11) and (12). Consequently, the loss function outputs larger values when there is a more significant discrepancy between the ground truth and the classification result. In addition, to ensure a certain level of accuracy in the attention edges and nodes, the sum of the loss functions $L_{\mathrm{per}}$ and $L_{\mathrm{att}}$ at the attention branch point is minimized. By minimizing the defined loss function $L_{\mathrm{total}}$, the parameters in the STA-GCN are determined.

## 3. Experimental Results

In this section, we present the experimental results to evaluate the classification performance of the proposed method. This experiment classifies expert–novice levels using motion data during sports activities, i.e., soccer and diving. In addition, we quantitatively evaluated the classification performance and discussed the effectiveness of explaining the classification results by visualizing the attention nodes.

### 3.1. Experimental Settings

In this subsection, we explain the experimental setting. To evaluate the classification performance of our GCN-based method, we used the expert–novice soccer dataset [47] and the action quality assessment (AQA) dataset [48]. These datasets contain motion data on sports and their expert–novice levels. Specifically, the expert–novice soccer dataset contains motion data of eight participants for nine types of soccer plays (penalty kick (PK), free kick (FK), direct shot (DS), cross shot (CS), volley, long dribble, straight dribble, short dribble, and juggling), four times each, for 288 samples. These motion data were obtained using the PERCEPTION NEURON PRO (https://neuronmocap.com), which was used to capture whole-body motion [49,50]. The nine types of soccer plays in this dataset are illustrated in Figure 5. In this dataset, the number of motion data frames differs based on the participant and the specific play. Note that the proposed method requires the number of frames in the input motion data to be identical. Therefore, to unify the temporal duration of all motion data, downsampling was performed by sampling the data at regular intervals to match the shortest motion data. In addition, each soccer play was classified according to a four-tiered expert–novice level, predetermined by individuals with more than five years of soccer experience. The AQA dataset consists of videos of athletes from seven sports (e.g., diving and 10 m platform) taken during the summer and winter Olympics. With this dataset, the experiment used motion data extracted from videos via MediaPipe 2 (https://google.github.io/mediapipe/, accessed on 23 February 2024). Because of the challenges of capturing motion data from the complex movements and changing camera angles present in many of the videos in the AQA dataset, only the “10 m platform single dive” data were used. The 10 m platform single dive in the AQA dataset is illustrated in Figure 6. The AQA dataset shows variability in motion data acquisition time across athletes and actions, which is similar to the expert–novice dataset. Therefore, to unify the temporal duration of all motion data, downsampling was performed in the same manner as in the expert–novice dataset experiment. The obtained motion data comprised 367 samples, of which 321 samples were used as training data and the remaining samples as test data. Each sample was given a score between 21.60 and 102.60 points. In the experiment, samples were categorized into four expertise levels, ranging from novice to expert, on the basis of the quartiles derived from their scores. To evaluate classification performance, we used the mean absolute error (MAE) and accuracy, which are defined as follows:(14)$$\begin{array}{c} \mathrm{MAE}=\frac{1}{K}\sum_{k=1}^{K}\left|g_{k}-r_{k}\right|, \end{array}$$
(15)$$\begin{array}{c} \mathrm{Accuracy}=\frac{\mathrm{Number}\;\mathrm{of}\;\mathrm{correctly}\;\mathrm{classified}\;\mathrm{samples}}{\mathrm{Number}\;\mathrm{of}\;\mathrm{all}\;\mathrm{samples}}. \end{array}$$

In Equation (Equation 14), $g_{k}$ and $r_{k}$ (k=1,2,⋯,K;K representing the number of test samples) denote the ground truth and classification results for the *k*-th test sample, respectively. In the MAE equation, $|g_{k}-r_{k}|$ denotes the difference between the actual expert–novice levels and the classification results. Therefore, the MAE calculated from Equation (Equation 14) indicates the extent to which the classification results deviate from the actual expert–novice level. A lower MAE indicates smaller classification errors, whereas a higher accuracy indicates a larger number of samples in which the classification results match the actual expert–novice level.

To evaluate the classification performance of the proposed method (PM), it was compared with the following eight comparative methods: the ST-GCN [29], ST-GCN with the proposed loss function $L_{\mathrm{total}}$ (ST-GCN w/$L_{\mathrm{total}}$), STA-GCN [33], STA-GCN with $L_{\mathrm{total}}$ (STA-GCN w/$L_{\mathrm{total}}$), ConfSTA-GCN [41], ConfSTA-GCN with $L_{\mathrm{total}}$ (ConfSTA-GCN w/$L_{\mathrm{total}}$), spatiotemporal graph ConvNeXt (TSGCNeXt) [51], and proposed method without $L_{\mathrm{total}}$ (PM w/o $L_{\mathrm{total}}$). The ST-GCN, which is capable of incorporating spatial and temporal information, is GCN-based. As a basic method in GCN-based classification that considers spatiotemporal information, the ST-GCN was used as a comparative method. The STA-GCN is a GCN-based classification method that introduces the conventional attention mechanism. The ConfSTA-GCN verifies the effectiveness of the computation of confidence measures in the PM. The ST-GCN w/$L_{\mathrm{total}}$, STA-GCN w/$L_{\mathrm{total}}$, ConfSTA-GCN w/$L_{\mathrm{total}}$, and PM w/o $L_{\mathrm{total}}$ were used to verify the effectiveness of our loss function $L_{\mathrm{total}}$. TSGCNeXt is a state-of-the-art method for GCN-based classification using motion data.

In this experiment, the number of joints in the expert–novice soccer and AQA datasets is 22 and 33, respectively. For the proposed and comparative methods, the learning rate and the batch size were set to 0.01 and 64, respectively, and this experiment used stochastic gradient descent [29] as the optimization approach. In addition, the kernel size of T-GC was set to nine, consistent with the conditions of [29,39].

### 3.2. Evaluation of Expert–Novice Level Classification Performance

This subsection shows the performance of expert–novice level classification via the proposed and comparative methods. Table 1 and Table 2 show the MAE and accuracy of the expert–novice level classification results obtained using the proposed and comparative methods for the expert–novice soccer dataset. Furthermore, the MAE and accuracy of the classification results for the AQA dataset are presented in Table 3. From these performance indices, the PM outperforms all comparative methods, demonstrating its effectiveness. Because the PM outperforms the ST-GCN and ST-GCN w/$L_{\mathrm{total}}$, we can conclude that it can classify with higher accuracy than the ST-GCN when an attention mechanism is introduced. Furthermore, because our method outperforms the STA-GCN and STA-GCN w/$L_{\mathrm{total}}$, accurate classification becomes feasible using the confidence-aware attention mechanism. By comparing the classification results of the PM, ConfSTA-GCN, and ConfSTA-GCN w/$L_{\mathrm{total}}$, we can verify the effectiveness of the node-level attention mechanism that can control the impact of each attention node. Because the proposed method outperforms the PM w/o $L_{\mathrm{total}}$, we can confirm the effectiveness of the expert–novice level classification using the loss function $L_{\mathrm{total}}$. Finally, by comparing the classification results of the PM and TGCNeXt, we confirm that the proposed method outperforms the state-of-the-art method for GCN-based classification using motion data. These results confirm that the PM allows for accurate classification of expert–novice levels by employing the confidence-aware node-level attention mechanism and the loss function $L_{\mathrm{total}}$.

In addition, the confusion matrices for the PM, PM w/o $L_{\mathrm{total}}$, and ConfSTA-GCN w/$L_{\mathrm{total}}$ are shown in Figure 7 and Figure 8. These results demonstrate that the proposed method is capable of accurately classifying the expert–novice levels compared with the PM w/o $L_{\mathrm{total}}$ and ConfSTA-GCN w/$L_{\mathrm{total}}$ and that can classify the expert–novice levels closely to the ground truth.

Examples of the classification results of the PM and ConfSTA-GCN for the expert–novice soccer and AQA datasets are shown in Figure 9 and Figure 10, respectively. These results confirm that by considering the ordinality between the expert and novice levels, we can accurately classify the expert–novice levels and enable classifications that are close to the ground truth. Consequently, in GCN-based classification, we verify the effectiveness of the confidence-aware node-level attention mechanism and the importance of considering the ordinality of the expert–novice levels.

This evaluation of expert–novice level classification performance demonstrates an improvement in classification performance, attributable to the contributions of this study, which include enhancements to the existing approach (ConfSTA-GCN) and the construction of a classification model that considers the ordinality between classes. Specifically, the effectiveness of improvements to the existing approach was verified by comparing the PM and ConfSTA-GCN w/$L_{\mathrm{total}}$. Furthermore, the efficacy of the classification model that considers the ordinality between classes was confirmed through the comparison of the PM and PM w/o $L_{\mathrm{total}}$. Consequently, this study achieves the research objectives of enhancing the existing approach and improving the classification performance of expert–novice level classification through a model that accounts for the ordinal relationships between classes.

### 3.3. Visualization Results of Attention Nodes

This subsection shows the visualization results of the attention nodes and discusses the effectiveness of the PM. Figure 11, Figure 12 and Figure 13 show examples of the visualization of the attention nodes for the PK, FK, and DS categories in the expert–novice soccer dataset. The frames visualized were selected as the frames with the largest standard deviation between attention nodes. Furthermore, Figure 14 shows an example of the visualization of the attention nodes in the AQA dataset regarding the frames when the standard deviation of the attention nodes is maximum. Figure 11, Figure 12 and Figure 13 show that participants with lower expert–novice levels are confirmed to shoot using only their legs.

Specifically, the visualization results indicate that participants with lower expert–novice levels have higher values in the nodes associated with the lower body. Conversely, participants with higher expert–novice levels are observed to effectively use their upper body when shooting. Figure 14 demonstrates that expert levels of expertise have higher values of attention nodes in the head and shoulder.

Figure 15, Figure 16, Figure 17 and Figure 18 focus on depicting the average values of attention nodes across all frames for each sample, confirming an overview of the trends in the whole sample. A comparison between attention nodes in Figure 11, Figure 12, Figure 13 and Figure 14 and averaged attention nodes reveal that in PK, FK, and DS categories, participants with lower expert–novice levels confirm higher values in nodes associated with the lower body. Conversely, it can be confirmed that participants with higher expert–novice levels exhibit higher values in nodes related to the upper body. This result is consistent with the results confirmed by the visualization results in frames with the highest standard deviation across both datasets. These results confirm that the visualization outcomes of attention nodes in this experiment are independent of the frame.

Figure 19, Figure 20, Figure 21 and Figure 22 depict the visualization of averaged attention nodes across all frames for each action at expert–novice levels, allowing for a comparison of how attention nodes between expert–novice levels. Despite the varying number of participants in each level, the values of attention nodes for the PK, FK, DS, and the AQA dataset exhibit similar trends to those confirmed in Figure 11, Figure 12, Figure 13 and Figure 14 and Figure 15, Figure 16, Figure 17 and Figure 18. In the expert–novice level soccer dataset, participants with higher expert–novice levels show elevated values in nodes associated with the upper body. In the AQA dataset, samples from expert participants show high values in nodes related to the upper body and foot. These results are corroborated by quantitative results from each dataset, confirming that the visualized attention nodes contribute to the classification process.

These results suggest that the visualization approach proposed consistently captures the importance of soccer-specific movements across different frame counts and samples, highlighting their relevance to varying expert–novice levels. Moreover, it successfully identifies the significance of movements specific to diving and swimming, demonstrating their relevance to expert–novice levels.

## 4. Conclusions

In this study, we proposed a method for classifying expert–novice levels using motion data via a GCN that introduces a confidence-aware node-level attention mechanism. The PM effectively solves the problem of using unimportant features in existing methods. In particular, the PM calculates the probability of belonging to an actual expert–novice level when specific attention nodes are excluded, and the calculated probabilities are regarded as a confidence measure. Consequently, our method can compare the attention value of each node based on the confidence of the classification. This solves the attention mechanism problem and enables accurate classification. Furthermore, because the expert–novice levels have ordinalities, we construct a classification model that considers ordinalities, thereby improving classification performance.

Because of the constraint in the PM, the number of frames in the input motion data must be uniform, and downsampling is performed. However, there is a problem with downsampling, which can result in the lack of important frames for accurate classification. Therefore, constructing a model capable of handling motion data with different time lengths remains a challenge for future work.

## Figures and Tables

**Figure 1 sensors-24-03033-f001:**
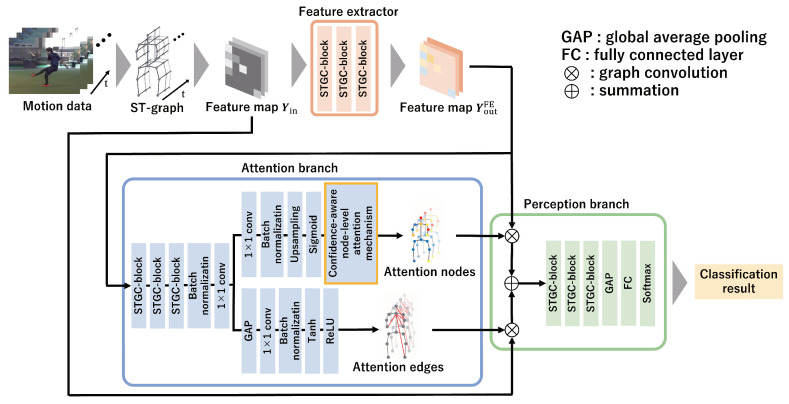
Overview of the proposed method. In the proposed method, feature maps extracted from graphed motion data are used to calculate attention nodes and edges through an attention mechanism that considers the confidence measure in emphasizing elements crucial for classification. Subsequently, the classification results of the expert–novice levels are obtained using the perception branch. Furthermore, the attention nodes used in the classification are visualized.

**Figure 2 sensors-24-03033-f002:**
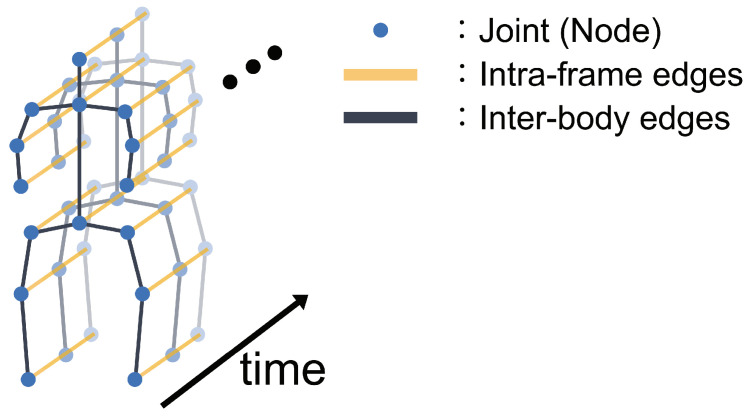
Overview of ST-graph constructed using the proposed method. The ST-graph is constructed by connecting nodes representing joints (blue points) with inter-frame edges (yellow lines) and intra-body edges (black lines).

**Figure 3 sensors-24-03033-f003:**
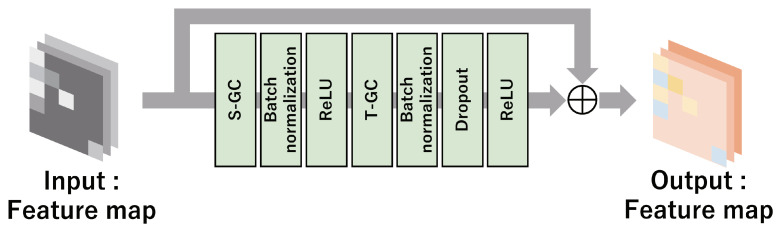
Network configuration of STGC-block in the proposed method.

**Figure 4 sensors-24-03033-f004:**
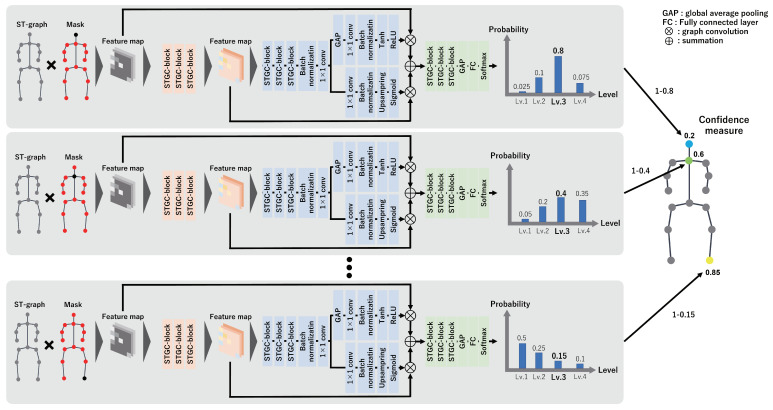
Overview of the calculation approach for the confidence measure. The proposed method calculates the probability of belonging to each expert–novice level by masking one attention node (setting its attention node to zero) and derives the confidence measure on the basis of the probability value. In this attention mechanism, the product of the calculated confidence measure and the attention node is taken, allowing the calculation of controlled attention nodes.

**Figure 5 sensors-24-03033-f005:**
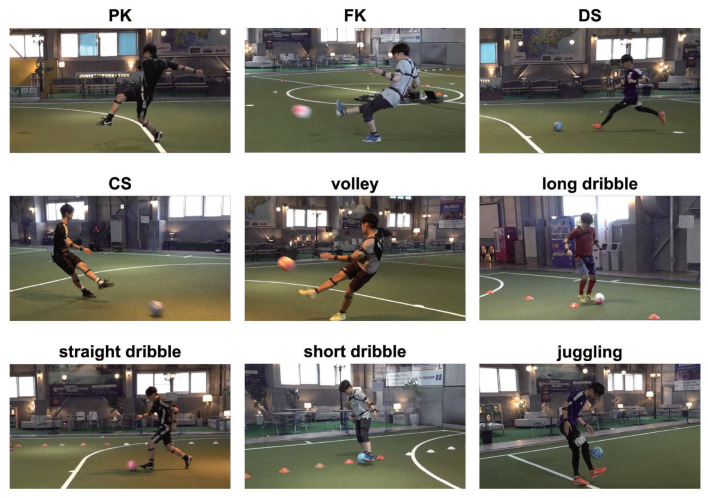
Nine types of soccer plays included in the expert–novice soccer dataset.

**Figure 6 sensors-24-03033-f006:**
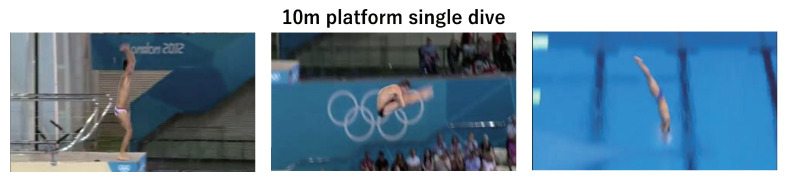
10 m platform single dive included in the AQA dataset.

**Figure 7 sensors-24-03033-f007:**
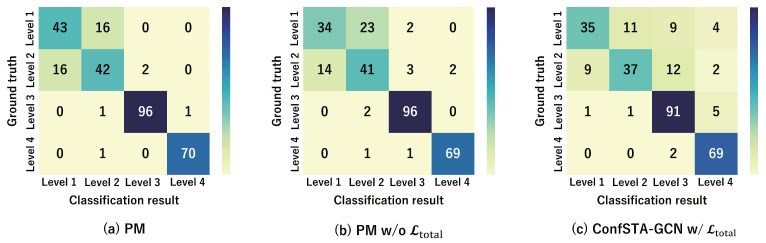
Confusion matrices of expert–novice level classification results obtained using the PM, PM w/o $L_{\mathrm{total}}$, and ConfSTA-GCN w/$L_{\mathrm{total}}$ for the expert–novice soccer dataset.

**Figure 8 sensors-24-03033-f008:**
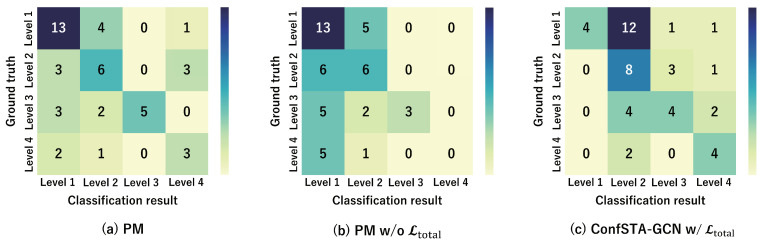
Confusion matrices of expert–novice level classification results obtained using the PM, PM w/o $L_{\mathrm{total}}$, and ConfSTA-GCN w/$L_{\mathrm{total}}$ for the AQA dataset.

**Figure 9 sensors-24-03033-f009:**
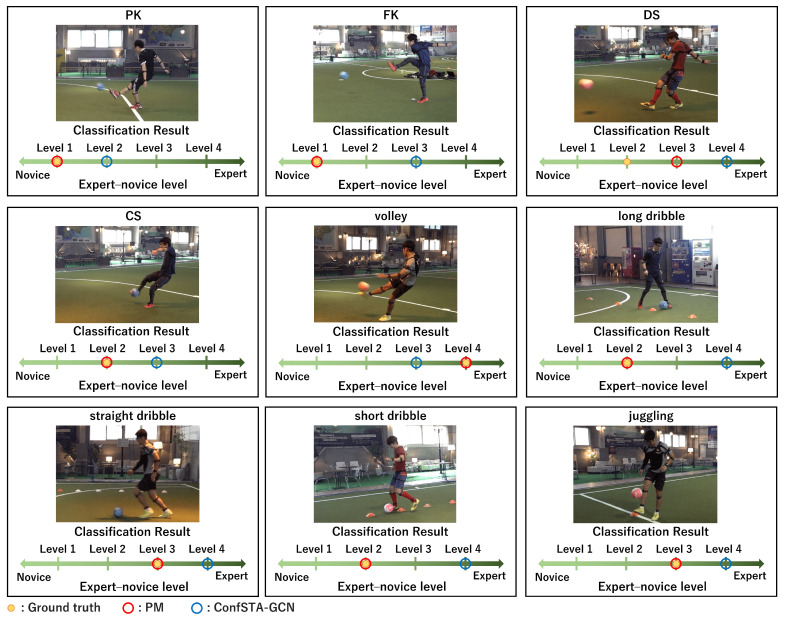
Examples of expert–novice level classification results obtained using the PM and previous method for the expert–novice soccer dataset.

**Figure 10 sensors-24-03033-f010:**
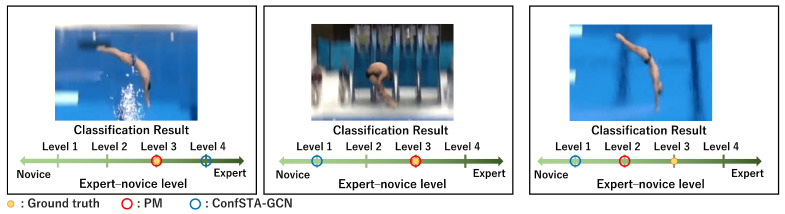
Examples of expert–novice level classification results obtained using the PM and previous method for the AQA dataset.

**Figure 11 sensors-24-03033-f011:**
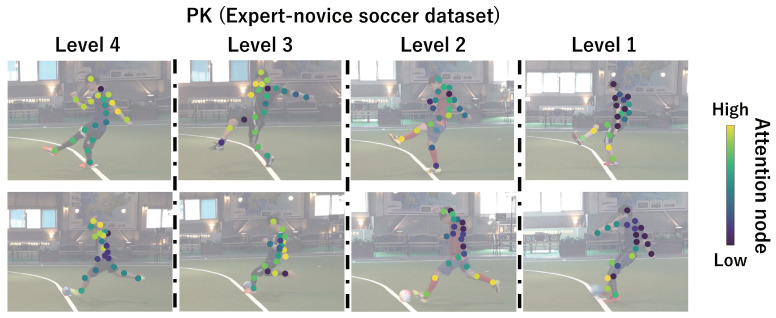
Examples of the visualization of attention nodes for penalty kick in the expert–novice soccer dataset using the PM.

**Figure 12 sensors-24-03033-f012:**
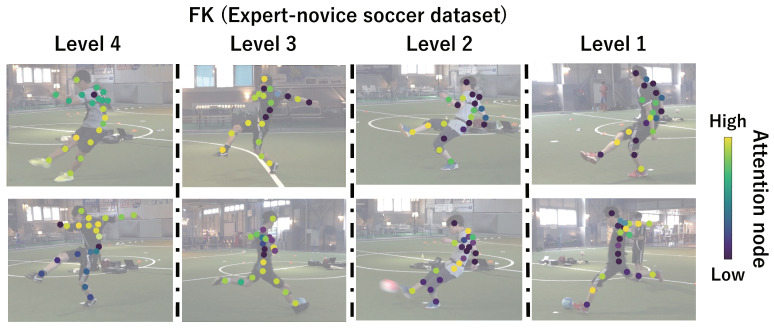
Examples of the visualization of attention nodes for free kick in the expert–novice soccer dataset using the PM.

**Figure 13 sensors-24-03033-f013:**
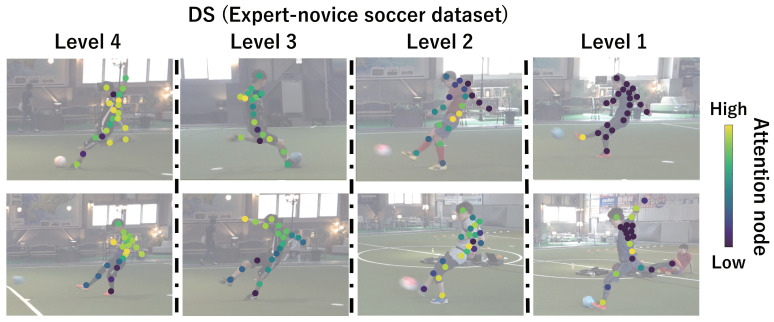
Examples of the visualization of attention nodes for direct shot in the expert–novice soccer dataset using the PM.

**Figure 14 sensors-24-03033-f014:**
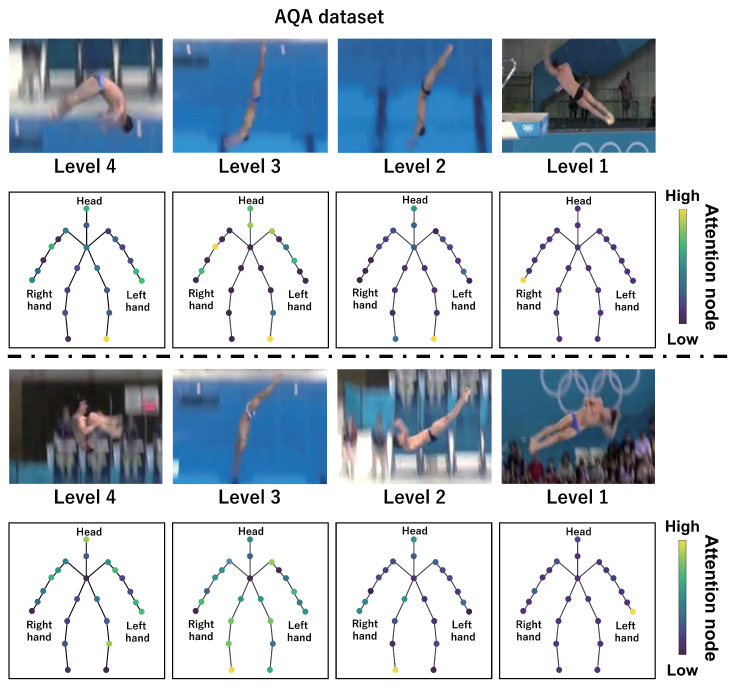
Examples of the visualization of attention nodes for the AQA dataset using the PM.

**Figure 15 sensors-24-03033-f015:**
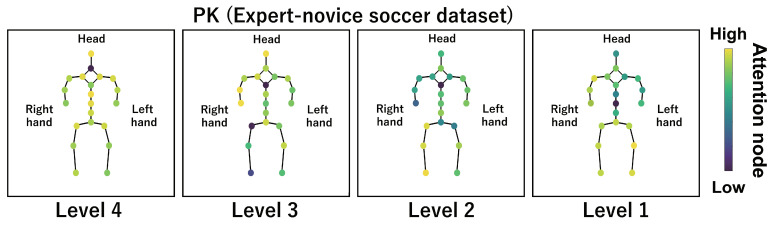
Examples of the visualization of attention nodes for the PK (expert–novice soccer dataset) using the PM. The attention nodes are averaged across all frames for each sample.

**Figure 16 sensors-24-03033-f016:**
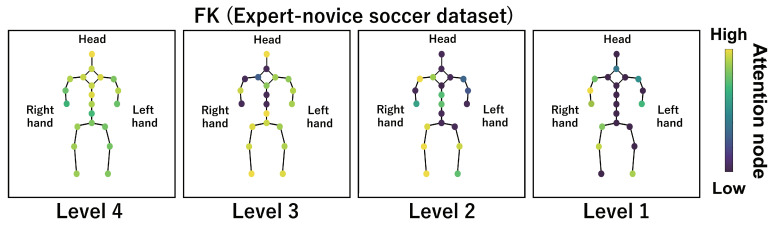
Examples of the visualization of attention nodes for the FK (expert–novice soccer dataset) using the PM. The attention nodes are averaged across all frames for each sample.

**Figure 17 sensors-24-03033-f017:**
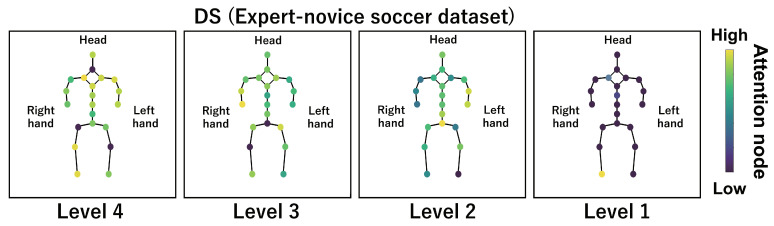
Examples of the visualization of attention nodes for the DS (expert–novice soccer dataset) using the PM. The attention nodes are averaged across all frames for each sample.

**Figure 18 sensors-24-03033-f018:**
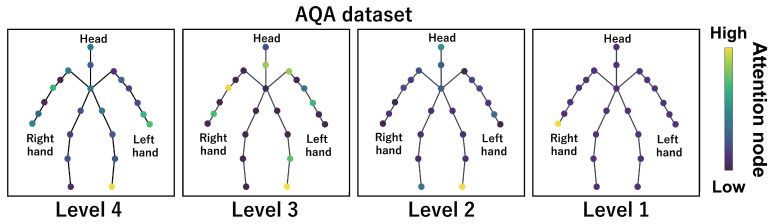
Examples of the visualization of attention nodes in the AQA dataset using the PM. The attention nodes are averaged across all frames for each sample.

**Figure 19 sensors-24-03033-f019:**
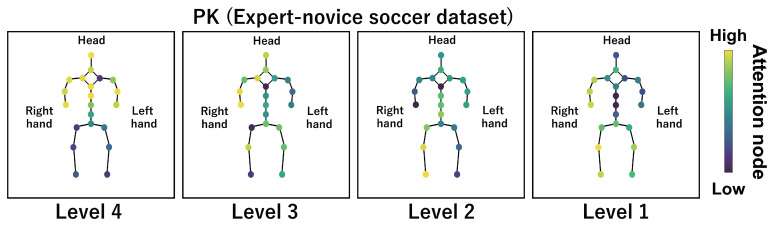
Examples of the visualization of attention nodes for the PK (expert–novice soccer dataset) using the PM. The attention nodes are averaged across all frames for each action by expert–novice levels.

**Figure 20 sensors-24-03033-f020:**
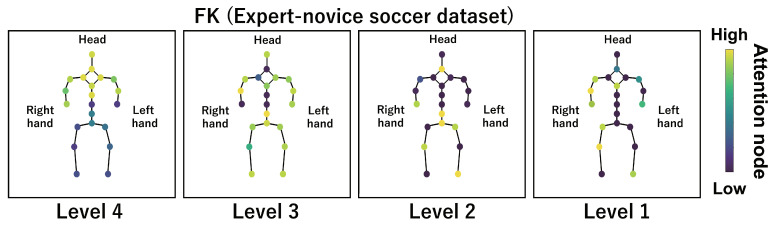
Examples of the visualization of attention nodes for the FK (expert–novice soccer dataset) using the PM. The attention nodes are averaged across all frames for each action by expert–novice levels.

**Figure 21 sensors-24-03033-f021:**
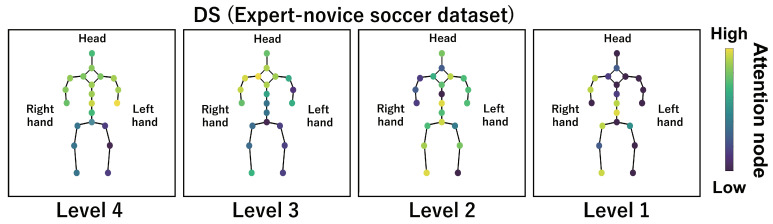
Examples of the visualization of attention nodes for the DS (expert–novice soccer dataset) using the PM. The attention nodes are averaged across all frames for each action by expert–novice levels.

**Figure 22 sensors-24-03033-f022:**
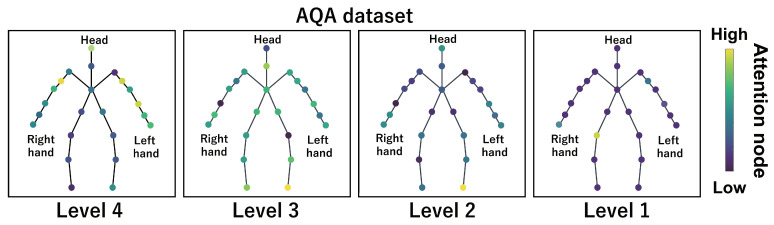
Examples of the visualization of attention nodes for AQA dataset using the PM. The attention nodes are averaged across all frames for each action by expert–novice levels.

**Table 1 sensors-24-03033-t001:** MAE (↓) of expert–novice level classification results obtained using the proposed and comparative methods for the expert–novice soccer dataset.

	ST-GCN [29]	ST-GCN w/$L_{\mathrm{total}}$	STA-GCN [39]	STA-GCN w/$L_{\mathrm{total}}$	ConfSTA-GCN [41]	ConfSTA-GCN w/$L_{\mathrm{total}}$	TGC NeXt [51]	PM w/o $L_{\mathrm{total}}$	PM
PK	0.844	0.375	0.594	0.313	0.281	0.344	0.563	0.250	**0.188**
FK	0.656	0.594	0.156	0.281	0.313	0.188	0.500	0.125	**0.0625**
DS	0.500	0.656	0.563	0.250	0.125	**0.0938**	1.16	**0.0938**	**0.0938**
CS	0.594	0.594	0.688	0.438	0.594	**0.188**	0.375	0.344	0.313
volley	0.594	0.688	0.250	0.219	0.250	**0.125**	0.656	0.188	0.156
long dribble	0.406	0.656	0.500	0.375	0.344	0.438	0.688	**0.0625**	**0.0625**
straight dribble	0.688	0.531	0.281	0.156	0.313	0.313	0.594	0.125	**0.0938**
short dribble	0.719	0.750	0.219	0.188	0.281	0.281	1.22	0.125	**0.0313**
juggling	0.531	0.438	0.563	0.375	0.531	0.406	0.906	0.344	**0.188**
Average	0.615	0.587	0.424	0.288	0.337	0.263	0.740	0.278	**0.132**

**Table 2 sensors-24-03033-t002:** Accuracy (↑) of expert–novice level classification results obtained using the proposed and comparative methods on expert–novice soccer dataset.

	ST-GCN [29]	ST-GCN w/$L_{\mathrm{total}}$	STA-GCN [39]	STA-GCN w/$L_{\mathrm{total}}$	ConfSTA-GCN [41]	ConfSTA-GCN w/$L_{\mathrm{total}}$	TGC NeXt [51]	PM w/o $L_{\mathrm{total}}$	PM
PK	0.469	0.719	0.656	**0.813**	0.750	0.750	0.563	0.750	**0.813**
FK	0.500	0.406	0.844	0.719	0.688	0.813	0.563	0.875	**0.938**
DS	0.594	0.500	0.781	0.813	0.875	0.906	0.344	**0.938**	0.906
CS	0.594	0.563	0.656	0.563	0.625	**0.813**	0.625	0.719	0.719
volley	0.688	0.688	0.750	0.781	0.813	**0.938**	0.500	0.813	0.844
long dribble	0.656	0.500	0.718	0.750	0.781	0.750	0.469	**0.938**	**0.938**
straight dribble	0.594	0.656	0.813	0.875	0.781	0.781	0.500	0.875	**0.906**
short dribble	0.656	0.563	0.875	0.875	0.844	0.813	0.375	0.875	**0.969**
juggling	0.531	0.563	0.688	0.656	0.594	0.688	0.406	0.719	**0.813**
Average	0.587	0.573	0.753	0.760	0.750	0.778	0.483	0.833	**0.872**

**Table 3 sensors-24-03033-t003:** MAE and accuracy of expert–novice level classification results obtained using the proposed and comparative methods for the AQA dataset.

	MAE (↓)	Accuracy (↑)
ST-GCN [29]	1.17	0.348
ST-GCN w/$L_{\mathrm{total}}$	1.13	0.348
STA-GCN [39]	1.00	0.348
STA-GCN w/$L_{\mathrm{total}}$	0.935	0.391
ConfSTA-GCN [41]	0.913	0.370
ConfSTA-GCN w/$L_{\mathrm{total}}$	0.891	0.413
TGCNeXt [51]	1.09	0.391
PM w/o $L_{\mathrm{total}}$	0.870	0.478
PM	**0.696**	**0.587**

## Data Availability

Publicly available datasets were analyzed in this study. The public datasets used in our experiment are available at https://github.com/LMD-datasets/Expert-NoviceSoccerDataset (accessed on 5 March 2024) and http://rtis.oit.unlv.edu/datasets.html (accessed on 5 March 2024).

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
