# Peer review of "Expert–Novice Level Classification Using Graph Convolutional Network Introducing Confidence-Aware Node-Level Attention Mechanism"

_sensors, 2024, doi:10.3390/s24103033_

Round 1

Reviewer 1 Report

Comments and Suggestions for Authors

This research deals with the problem of expert-novice level classification for motion data, and the objective is to realize a discriminator that can analyze the reasons for identifying results while maintaining high identification accuracy. In particular, this paper proposes a method to improve the classification performance by utilizing an ordered 4-class classification instead of the conventional 2-class classification. Traditionally, graph convolutional networks (GCN) have been used for expert-novice level classification tasks, but the authors consider GCN a method that does not explain the identification results. In this paper, the authors introduce a spatiotemporal graph called a time frame, in which nodes are joined in the temporal direction and nodes are connected by edges. The authors attempt to ensure the explanatory power of the identification results by introducing a time frame, a spatiotemporal graph in which nodes are connected by edges.    

Expert-novice level classification methods for sports videos, such as the one used in this paper, are generally handled as time-series data, and they are also widely used in combination with attention maps to improve discrimination accuracy. In this respect, it can be said that widely used methods are being used in response to research trends. The first is the masking method for confidence measurement, and the second is that the number of confidence classes is set to four. The improvement of estimation accuracy in expert-novice level classification tasks is a very common evaluation measure in this field, and evaluation experiments have been conducted by applying it to standardized evaluations.    

Therefore, the authors have proposed a graph convolutional network with spatiotemporal connections and a graph convolutional network that can acquire the degree of attention to each node by learning (STA-GCA). This method has been previously presented at international conferences. This paper shows that classification accuracy can be improved by solving a 4-class classification task by ordering the number of classes to identify the degree of expert-novice level instead of the 2-class classification of "advanced and novice." In particular, to calculate which node dominantly contributes to the estimation of the expert-novice level, a masking operation is performed to set the contribution rate of the target node to 0, and the complement of the "class classification probability in the absence of the contribution of the node" (Equation.5) is obtained. Figure.4), which is highly useful. It has been shown that this method provides not only the classification result but also an attention distribution based on the confidence distribution of the nodes that dominate the classification result (Figure.11-13).  

The visualization of the reliability of each node and the improvement of the accuracy of the classification is highly useful. For the soccer dataset, MAE, precision, and confusion matrices are shown (Table.1-2, Figure.7), and the visualization of attention nodes based on their confidence level is shown (Figure.11-13). On the other hand, for the AQA dataset, MAE, precision, and confusion matrices are shown (Table.3,Figure.8), but the visualization results of attention nodes based on the confidence level are not shown. If visualization results of attention nodes based on reliability exist for the AQA dataset, they should be shown the same as Figure.11-13 to be consistent with the claim. Conversely, if there is some reason why the visualization of attention nodes in the AQA dataset could not be done, it should be shown in this paper, and the conclusion should be revised.  

The results for confidence-based attention nodes are shown in Figures.11-13, which can be read as having frame-dependent values, being the f-th frame-specific attention node as shown in Equation (5). Indeed, evidence is presented for Figures 11-13 as a visualization of the attention node, and it is explained that the attention node identified as a novice is a toe, as shown on the right of Figure 13. However, I cannot read from the data presented in this paper whether the attention node is maintained in frames other than the f-th frame, and we cannot deny the possibility that the author has arbitrarily selected frames that are easy to explain for their convenience. If the correlation between expert-novice level and attention nodes is obtained, the distribution of attention nodes is expected to be reproduced in an expert-novice-dependent manner. However, only a few samples between expert-novice level and attention nodes are shown. Moreover, the reproducibility between frames of attention nodes is not mentioned. In addition, the reproducibility between the expert-novice level of the attention nodes is not explained. Therefore, it cannot be said that whether the visualization of attention nodes is possible is sufficiently answered.  

1) The source of the open source dataset is disclosed, and there is no doubt about it. 2) The source of the use of the open-source dataset is disclosed, and there is no doubt about it. 3) "Firs" at the end of line 125 seems a typo. 4) The expert-novice levels are described as Level 4 to Level 1 between Figure 7 and Figure 13. It was read from the text that "Level 4 is the expert level, and Level 1 is the novice level. However, it did not seem explicitly stated in the text, so adding a caption such as Level 4 (Expert), Level 1 (Novice), etc, would be better. I recommend that you use the same notation as in the previous study by the authors [34] Fig. 3.

Comments on the Quality of English Language

It is advisable to have your manuscript proofread by a native English speaker.

Reviewer 2 Report

Comments and Suggestions for Authors

1. The announced contributions and research questions mentioned in the introduction should be thoroughly demonstrated and discussed in other relevant sections of the manuscript. It is essential to clearly present the evidence supporting the announced contributions.

2. The literature survey should encompass recent studies related to the subject domain, with a specific focus on expert-novice level classification. This will strengthen the relevance and credibility of the manuscript.

3. The methodology lacks an introduction explaining its purpose. It is recommended that an introductory statement be provided to clarify the focus of this paper.

Comments on the Quality of English Language

Null

Round 2

Reviewer 2 Report

Comments and Suggestions for Authors

The current version is already satisfactory.